# Influence of the Porosity of Polymer Foams on the Performances of Capacitive Flexible Pressure Sensors [note 1]

**DOI:** 10.3390/s19091968

**Published:** 2019-04-26

**Authors:** Sylvie Bilent, Thi Hong Nhung Dinh, Emile Martincic, Pierre-Yves Joubert

**Affiliations:** Centre de Nanosciences et de Nanotechnologies, CNRS, Univ. Paris-Sud, Université Paris-Saclay, 91120 Palaiseau, France; sylvie.bilent@c2n.upsaclay.fr (S.B.); thi-hong-nhung.dinh@c2n.upsaclay.fr (T.H.N.D.); emile.martincic@c2n.upsaclay.fr (E.M.)

**Keywords:** polymer-based flexible pressure sensors, microporous PDMS foam, electromechanical characterizations, sensor behavioral modeling, sensor sensitivity and measurement range

## Abstract

This paper reports on the study of microporous polydimethylsiloxane (PDMS) foams as a highly deformable dielectric material used in the composition of flexible capacitive pressure sensors dedicated to wearable use. A fabrication process allowing the porosity of the foams to be adjusted was proposed and the fabricated foams were characterized. Then, elementary capacitive pressure sensors (15 × 15 mm^2^ square shaped electrodes) were elaborated with fabricated foams (5 mm or 10 mm thick) and were electromechanically characterized. Since the sensor responses under load are strongly non-linear, a behavioral non-linear model (first order exponential) was proposed, adjusted to the experimental data, and used to objectively estimate the sensor performances in terms of sensitivity and measurement range. The main conclusions of this study are that the porosity of the PDMS foams can be adjusted through the sugar:PDMS volume ratio and the size of sugar crystals used to fabricate the foams. Additionally, the porosity of the foams significantly modified the sensor performances. Indeed, compared to bulk PDMS sensors of the same size, the sensitivity of porous PDMS sensors could be multiplied by a factor up to 100 (the sensitivity is 0.14 %.kPa^−1^ for a bulk PDMS sensor and up to 13.7 %.kPa^−1^ for a porous PDMS sensor of the same dimensions), while the measurement range was reduced from a factor of 2 to 3 (from 594 kPa for a bulk PDMS sensor down to between 255 and 177 kPa for a PDMS foam sensor of the same dimensions, according to the porosity). This study opens the way to the design and fabrication of wearable flexible pressure sensors with adjustable performances through the control of the porosity of the fabricated PDMS foams.

## 1. Introduction

Flexible pressure sensors have raised strong interest in non-invasive health monitoring applications [1]. Indeed, they allow wearable devices to be developed and have numerous monitoring applications such as plantar support monitoring [2,3], wrist radial artery pulse wave monitoring [4], or bandage compression control in the therapy of chronic venous disorders [5]. Several flexible pressure sensing technologies are available [1] such as piezoresistive, piezoelectric, and capacitive sensors [6,7,8,9]. In this paper, we focused on capacitive pressure sensors because of their low sensitivity to temperature changes and the low power consumption readout electronics that can be associated with them. The basic principle of such sensors is to sense the deformation of a flexible polymer under stress by means of the capacitance changes read out between the electrodes integrated within the polymer, that move relatively to one another when a load is applied. Since these sensors are intended to be set in contact with the skin, they must be disposable and thus have a low production cost. To meet these requirements, silicone based polymers such as polydimethylsiloxane (PDMS) constitute good candidates for elaborating such sensors. Indeed, PDMS is a non-toxic material, which features dielectric properties as well as relevant mechanical properties, allowing up to 100% deformation without damage [10]. However, PDMS features highly non-linear deformation behavior and is hardly compressible [10]. These drawbacks actually limit the performances of PDMS-based capacitive sensors. Indeed, they enable a large pressure measurement range to be reached (over a 400 kPa pressure range [2,7,11]); however, the sensitivity is rather low (in the order of 1%.kPa^−1^ for pressure <10 kPa, and <0.05%.kPa^−1^ for higher pressure [7,12]). For this reason, some authors have developed patterned PDMS films in order to increase the sensor sensitivity. Such patterned films enable higher deformations of the sensor dielectric layer, hence allowing significantly higher sensitivities to be obtained (e.g., up to 55%.kPa^−1^ [13,14]). However, these sensors are actually operative in low measurement ranges (<20 kPa) where they exhibit strong non-linear behavior, in addition to being possibly more complex to fabricate and feature hardly predictable performances [15]. For these reasons, easy to design sensors featuring enhanced sensitivity in larger pressure ranges would be welcome to address applications related to human motion monitoring applications [9,14] including plantar pressure monitoring which require pressure ranges up to 300 kPa [7]. 

The aim of this study was to consider capacitive pressure sensors fabricated with PDMS foams, so that high sensitivities could be reached thanks to the introduction of microporosities within the polymer without drastically reducing the resulting measurement range. The secondary motivation was to determine a relationship between the porosity of the fabricated PDMS foams and the sensor performances, so that PDMS foam based sensors could be designed by adjusting the porosity of the used foams. PDMS foams can be obtained with a simple fabrication process by mixing the polymer with sugar particles and further dissolving them [16,17,18] to make microporosities appear within the material. In this paper, we proposed a methodology allowing PDMS foams with adjusted porosity to be fabricated, and we studied the influence of the obtained porosity on the mechanical performances of the resulting capacitive sensors. Furthermore, since polymer-based pressure sensors feature strongly non-linear behaviors, we proposed a behavioral model that allows for a wide variety of sensors to be objectively and quantitatively compared in terms of sensitivity and pressure range.

## 2. Sensor Principle

In this paper, we considered an elementary capacitive normal pressure sensor used as a work prototype. This was composed of two square shaped brass electrodes (*l* = 15 mm side length, 1 mm thickness) placed at each face of a microporous PDMS foam used as a compressible dielectric layer (see Figure 1). The capacitance *C* of the sensor is given by:(1)$$C=\frac{\varepsilon_{0}\varepsilon_{r}A}{d}$$ where *A* is the area of the electrodes; *ε_r_* is the relative permittivity of the foam; and *d* is the distance between the electrodes. When a load is applied to the sensor (Figure 1), the foam undergoes a mechanical compression resulting from the deformation of the polymer and the reduction of the size of the pores. As a result, the sensor capacitance increases because of both the reduction of the distance *d* and the increase of the foam permittivity. Indeed, the value of the foam permittivity results from a combination of the intrinsic permittivity of the constitutive polymer and the permittivity of the volume of air trapped within the foam pores (see Section 3.2). As a result, the foam permittivity is increased when the size of the pores is reduced by compression, since the proportion of air within the foam is reduced. Finally, the capacitance changes under load is given by $\Delta C=C-C_{0}$ where *C*_0_ is the load-free capacitance of the sensor given for the initial distance *d* = *d*_0_, and the initial value of *ε_r_*(*d*_0_) of the uncompressed foam.

## 3. PDMS Foam

### 3.1. Fabrication Process

Figure 2 shows the fabrication process of the microporous PDMS. PDMS is formed by mixing two components: silicon elastomer (Sylgard 184A) and a curing agent (Sylgard 184B) at a standard ratio of 10:1. The silicon elastomer and curing agent are mixed and placed in a desiccator for 60 min to degas. Household powder sucrose (C_12_H_22_O_11_) crystals were then added to the PDMS in volume ratios ranging from 4 to 6 (sugar powder) per 1 (PDMS), both volumes being estimated using measuring containers. The preparation was poured into a Petri dish and compacted to form a block of the mixture. The mixture was cured on a stove for 2 h at 90 °C. After curing, the mixture was removed from the Petri dish and soaked in a water bath to dissolve the sacrificial sugar crystals, thereby obtaining the microporous PDMS foam. Once the sugar was completely dissolved, the PDMS foam was again placed on the stove for 1 h 30 at 75 °C to remove the excess water. Finally, the microporous PDMS was cut to the shape of the electrodes to constitute the capacitive elementary sensor used for characterization purposes.

### 3.2. Characterization of the PDMS Foams

In order to study the influence of the porosity on the mechanical performances of the fabricated sensors, we first adjusted the size of the used sugar crystals, and then adjusted the sugar:PDMS ratio volume used to fabricate the foams. To do so, commercially available sugar crystals of unknown dimensions were sorted into three sets by means of 500 µm, 1000 µm, and 2000 µm sieving grids. The size distribution of the crystal within each set was estimated by evaluating the cross section of sugar particles picked up within each of the three sets, and observed by means of a Keyence VHX-1000 optical microscope (around 100 crystals were observed in each set). An example of sugar particles of small size is presented in Figure 3a.

The sizes of the used crystals, defined as the root mean square value of the observed crystal cross sections, were estimated from the distributions of the cross sections of the crystals, presented in Figure 3b. As a result, the crystal sizes were found to be 470 ± 100 µm (small size), 700 ± 230 µm (medium size), and 1100 ± 330 µm (large size). Furthermore, the cross sections of PDMS foam samples were observed by means of a scanning electron microscope (Hitachi S3600N). Figure 3c–e shows some examples of the pore distributions in foams fabricated with small, medium, and large crystals, respectively. The sizes of the obtained pores were rather difficult to accurately evaluate, since the orientation of the three-dimensional pores are actually unknown in this two-dimensional observation. Nevertheless, it can be qualitatively assessed that the sizes of the obtained pores correlated to the sizes of the sacrificial sugar crystals used to fabricate the pores. As a result, the fabricated 4:1 and 6:1 foams are referred as being obtained with either “small”, “medium”, or “large” pores throughout the rest of this paper. 

The porosity *f* of the fabricated foams is calculated from the density of the PDMS foam (*ρ_foam_*) and the bulk PDMS density (*ρ_bulk_*) using Equation (2): (2)$$f=1-\frac{\rho_{foam}}{\rho_{bulk}},$$ where *ρ_foam_* is determined from the ratio of the mass to the volume of the fabricated foam, and where *ρ_bulk_* = 1100 kg/m^3^ [19]. The relative permittivity of the PDMS foam (*ε_r_*) is a combination of the relative permittivity of PDMS (*ε_PDMS_* = 2.69 [20]) and the relative permittivity of air (*ε_air_* = 1). The resulting foam relative permittivity can be theoretically estimated using [17]:(3)$$\varepsilon_{r}=\varepsilon_{air}f+(1-f)\varepsilon_{PDMS},$$ where *f* is the foam porosity. The porosity and the expected relative permittivity of the various PDMS foams fabricated in this study are shown in Table 1, according to the pore sizes and sugar:PDMS ratios. In addition, the permittivity of the PDMS foam samples were experimentally estimated through the value of the capacitance measured between two 2 mm thick copper sheet electrodes cut to the size of the PDMS samples (5 cm diameter disk) and placed on both faces of the PDMS foam. The capacitance was measured with a HP 4192A impedance analyzer operated at 1 MHz. For the considered samples, the measured capacitances were found to be in the pF range and the relative permittivity of the foams was subsequently determined using Equation (1).

As observed in Table 1, the porosity of the fabricated foams significantly increased with the sugar:PDMS ratio (e.g., from around 80% for 4:1 foams up to over 84% for 6:1 foams with medium size pores). To a lesser extent, the foam porosity also increased with the size of the pores (e.g., from around 82% up to 83.5% for small and large pores in 6:1 PDMS foams, respectively,). Finally, as expected from Equation (4), the permittivity of the foams decreased when the porosity of the foam increased. 

## 4. Sensor Electromechanical Characterizations

In this section, elementary pressure sensors were fabricated as described in Figure 2 with PDMS foams of various porosities and 1 mm thick brass electrodes were added on both sizes of the PDMS foams for electromechanical characterization purposes. The sensors were characterized in terms of sensitivity and measurement range thanks to a dedicated electromechanical test bench. 

### 4.1. Measurement Setup 

The used electromechanical test bench is described in Figure 4a and is composed of (i) an indenter applying normal stress to the sensor under test through an ISEL iMC-S8 robotic arm, (ii) a FUTEK FSH00105 110 N force sensor connected to a HP34401A voltmeter measuring the stress applied in a 0–488 kPa pressure range, and (iii) a HP4192A impedance analyzer measuring the sensor capacitance changes at a frequency of 1 MHz. All of the measurement setup was controlled by a computer operating under MATLAB and used for the acquisition, storage, and processing of the measurement data.

A typical sensor response curve obtained with a 6:1 small pore PDMS foam sensor, is presented in Figure 4b (blue dots). As commonly observed with polymer based capacitive pressure sensors, the obtained response curve is strongly non-linear, the sensitivity of the sensor is high for low stress values and exhibits a saturation behavior for large stress values [1,2,6,11,16,18]. In the case of polymer-foam based sensors, the sensor capacitance increases with the applied pressure under the combined effects of the reduction of the foam thickness and the increase of the foam permittivity (see Section 2). To illustrate these effects, the evolution of the foam permittivity with the applied pressure is presented in Figure 4c. The permittivity was estimated from the measured capacitance *C*, the experimental values of *d* and Equation (1). One can note that the estimated permittivity changed from the initial value of *ε_r_*(*d*_0_) = 1.3 up to *ε_r_*(*d*) ≈ 2.6 when the pressure exceeded 100 kPa; the permittivity of the foam reached a saturation value which was close to the permittivity of bulky PDMS (*ε_PDMS_* = 2.69), meaning that the pores had been significantly reduced under compression, i.e., the porosity *f* tended toward zero in Equation (3). The changes of *d* with the applied pressure are presented in Figure 4d. The distance *d* was estimated from the initial value of the unloaded sensor (*d*_0_) and the set positions of the robotic arm applying the load onto the sensor (Figure 4a). One can note that the thickness *d* tended to have a minimum value under high pressure loads. This is consistent with the fact that, when the pores have almost been completely “closed”, the mechanical behavior of PDMS foam tends to show the same mechanical behavior as bulk PDMS, which is that of a hardly compressible polymer which features a Poisson coefficient close to 0.49 [21].

### 4.2. Sensor Behavioral Model 

In order to objectively and quantitatively compare the sensor performances, we proposed building an electromechanical behavioral model of the pressure sensor. The model used in this study was a first order exponential model, expressed by: (4)$$\frac{\Delta C}{C_{0}}\left(P\right)=\frac{\Delta C_{max}}{C_{0}}\left(1-e^{-(P/P_{C)}}\right),$$ where *C*_0_ is the initial capacitance value of the unloaded sensor; *ΔC_max_* is the maximum capacitance change; *P* is the applied pressure; and *Pc* is the characteristic pressure at which the capacitance change reaches 63% of its maximum value (Figure 4b). The choice of such a model is justified by the behavior of the sensor response curves. Indeed, as mentioned in Section 4.1, the capacitance changes were expected to be high at low pressure levels, since it increased with the combined effects of the reduction of *d* and the increase of the foam permittivity *ε_r_*(*d*) (Figure 4). On the other hand, at high pressure levels, the capacitance change reached a maximum value due to the saturation of the permittivity (Figure 4b) as well as the incompressibility of the foam featuring “closed” pores. In order to measure the relevance of the model, response curves of a 4:1 small pore foam sensor were measured during a load cycle (load increasing from 0 to 450 kPa) and a release cycle (load decreasing from 450 kPa back to 0 kPa). The model was fitted to the experimental data by the numerical adjustment of *ΔC_max_* and *P_c_*. Figure 5 shows the experimental curves together with the fitted model curves. One can note a fair agreement between the experimental and model curves fitted with a R^2^ higher than 0.99. Figure 5 also shows the average model curve, obtained by averaging the load and release model curves. 

In this paper, we proposed that the sensitivity *S* of the sensor can be defined by the tangent to the origin of the sensor adjusted model curve, and that the pressure measurement range *PR* corresponded to 95% of the maximum capacitance change. These quantities were used as the sensor performance features to compare the different sensors, and are expressed by: (5)$$\begin{array}{c} S\left(\%.kPa^{-1}\right)=\frac{\Delta C_{max}}{C_{0}P_{c}}\\ PR\left(kPa\right)=3P_{c} \end{array}$$

### 4.3. Influence of the Foam (Sugar:PDMS) Ratio

First, the influence of the sugar:PDMS ratios on the sensor performances were studied. To do so, sensors fabricated with 15 × 15 mm^2^ brass electrodes and PDMS foams (Figure 2) featured by 4:1 and 6:1 ratios foams (small pores) were considered. The thickness of the foams was *d*_0_ = 5 mm ± 0.5 mm. The sensors were characterized using the test bench shown in Figure 4 and electromechanical responses were obtained using load and release cycles: loads increasing from 0 up to 488 kPa were first applied (load), then loads decreasing from 488 to 0 kPa were applied (release). In Figure 6, one can note that the response curves obtained for the PDMS foam based sensor featured much higher capacitance changes than sensors fabricated with bulk PDMS. This confirmed that higher porosities significantly increased the sensor deformation under the load and capacitance changes. Additionally, a hysteresis appeared between the load and release curves. For each sensor response, the hysteresis was estimated by differentiation between the areas measured under the load and the release curves. It is expressed in percentage comparatively to the load curve. The hysteresis has been found to be lower than 8.85% for bulk PDMS sensors, and lower than 6.5% for PDMS foam based sensors. If the hysteresis in all cases was inferior to 10%, we chose to apply the proposed model (Equation (5)) on a mean response curve obtained by averaging the load and release curves. The assessed sensitivity and pressure range obtained using the proposed model are presented in Table 2 for each tested sensor. It can be observed that the porosity increased the sensitivity *S* of the sensor from 6.50%.kPa^−1^ to 13.07%.kPa^−1^ for 4:1 and 6:1, respectively; moreover, these sensitivities became significantly higher than the sensitivity of the bulk PDMS sensor with the same dimensions (0.14%.kPa^−1^). On the other hand, the pressure range *PR* decreased from 594 kPa (bulk PDMS) to 243 kPa (4:1 PDMS foam) and 166 kPa (6:1 PDMS foam) when the porosity increased. 

### 4.4. Influence of the Foam Pore Sizes 

The influence of the size of the pores constituting the PDMS foams were studied in this section. The load and release curves of the sensors fabricated with small, medium, and large pores are presented in Figure 7a for the 4:1 foams and in Figure 7b for the 6:1 foams. In both figures, one can see that higher pore sizes increased the sensor capacitance changes under load. For all considered foams, it can be observed in Figure 8 that the general tendency for a given sugar:PDMS ratio is that the increase of the pore size induced the increase of the sensor sensitivity (e.g., for *S* = 6.50 %.kPa^−1^ up to 8.85%.kPa^−1^ for 4:1 small and large foams, respectively), while the pressure ranges remained rather unchanged. This general remark does not completely apply to the 6:1 PDMS foam with large pores. This can be attributed to the fact that the 6:1 large pore PDMS foam is rather brittle and, due to some loss of material, the dimensions of the used sample might not be exactly similar to those of the other samples, so that the performances are not completely comparable to the other sensors. 

### 4.5. Influence of the Foam Thickness 

In this section, sensors fabricated with 4:1 and 6:1 foams of 5 mm and 10 mm thicknesses (medium pores) were compared. The sensor electromechanical responses are presented in Figure 9. One can see that an increase in the thickness enhanced the sensor deformation under load, and resulted in a small increase in the sensor sensitivity, as presented in Table 3. For example, S increased from 13.7%.kPa^−1^ up to over 15.7%.kPa^−1^ when d_0_ increased from 5 mm to 10 mm, for medium pore 6:1 PDMS foams. 

### 4.6. Influence of the Dynamics of the Applied Load 

In this section, a 6:1 medium pore PDMS foam based sensor was characterized under load applied continuously or discontinuously during the load-and-release cycles. In the continuous mode, the applied pressure was continuously increased or decreased during the load-and-release cycle and the complete cycle including the acquisition of a 100 measurement points was achieved in eight minutes. In the discontinuous mode, the applied pressure was increased or decreased step by step with a 10 s pause between each step, so that the complete cycle was achieved in 25 min. The obtained load and release curves are presented in Figure 10, and the performances parameters are shown in Table 4. One can see that the performances were rather unchanged whichever way the load was applied or released. However, the observed hysteresis was reduced from 4.58% down to 2.92% when the load was more “slowly” applied, i.e., the material has more time to recover its shape when the load is released progressively. 

## 5. Conclusions

In this study, PDMS foams with different porosities were obtained by controlling the size and the concentration of sugar crystals mixed with PDMS in the fabrication process. The obtained foams were characterized in terms of porosity, and electromechanical characterizations of capacitive PDMS foam based pressure sensors were carried out to study the influence of the porosity on the sensor performances. In order to quantitatively compare the sensor performances, a non-linear behavioral model was proposed and adjusted to the sensor pressure response curves to estimate the sensor performances in terms of sensitivity and pressure range. The main conclusion of the study is that the porosity of the foam significantly increases the sensitivity of the sensor: it is multiplied by a factor of 100 when compared to the sensitivity of a bulk PDMS sensor of the same size and shape. Additionally, the pressure range of PDMS foam based sensors is reduced by a factor of 2 to 3 depending on the porosity when compared with bulk PDMS sensors. Future works will focus on the shear stress performances of PDMS foam based sensors, so that three axes of flexible pressure sensors can be considered. To do this, fully flexible sensors will be microfabricated with flexible thin copper electrodes by means of a transfer of film fabrication process [11]. Further works will also focus on the dynamic behavior of PDMS foam sensors according to the porosity and the robustness of such sensors with respect to intensive use. This study opens the way to the design and fabrication of advanced wearable pressure sensors, the performances of which can be adjusted to the application aim through the control of the porosity of the fabricated foam.

## Figures and Tables

**Figure 1 sensors-19-01968-f001:**
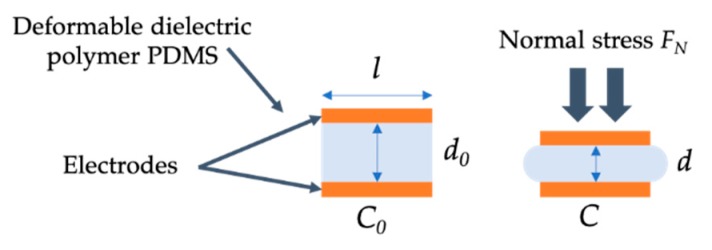
Principle of operation of the flexible normal pressure capacitive sensor.

**Figure 2 sensors-19-01968-f002:**
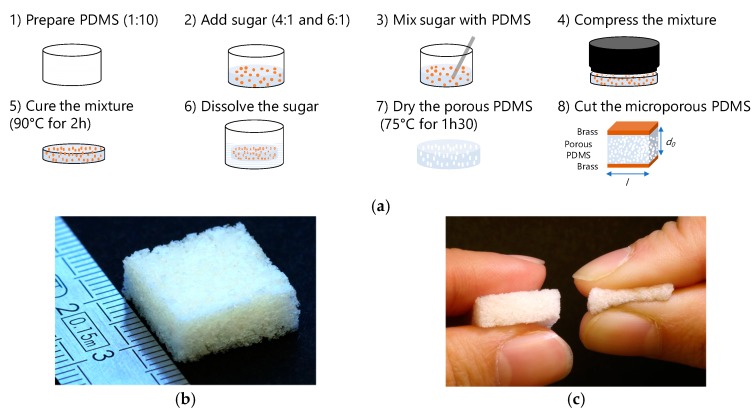
(**a**) Fabrication process of the microporous PDMS foams; (**b**) example of 4:1 small pore PDMS foam sample; (**c**) compressibility of the of 4:1 small pore PDMS foam sample.

**Figure 3 sensors-19-01968-f003:**
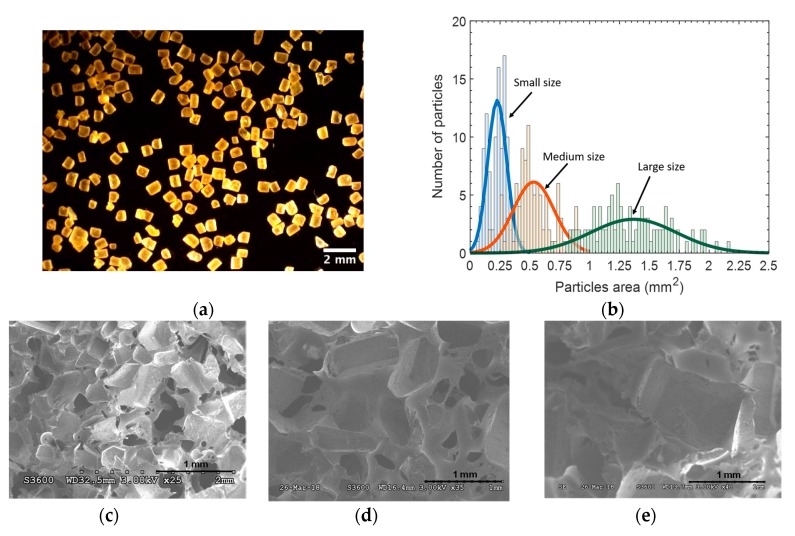
(**a**) Small sugar particles observed with an optical microscope; (**b**) Size distribution of sugar particles after the sieving procedure; (**c**–**e**) SEM images of the cross-section of microporous PDMS foams featuring small, medium, and large pores, respectively.

**Figure 4 sensors-19-01968-f004:**
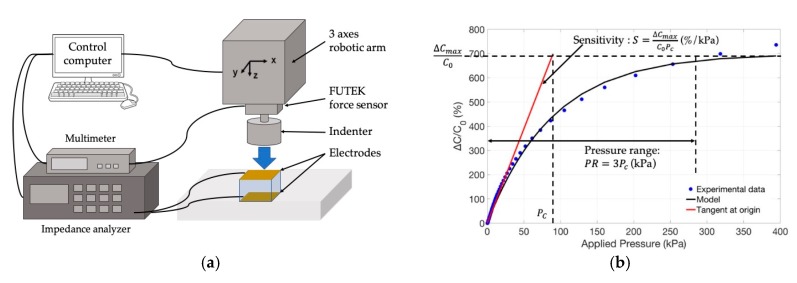
(**a**) Electromechanical test bench used to measure the sensor response curve (capacitance change versus applied pressure); (**b**) Typical sensor response curve (blue dots) for a sensor featuring a 6:1 small pore foam and first order exponential behavior model (black line) allowing sensor sensitivity (*S*) and pressure range (*PR*) to be defined and estimated from the characteristic pressure *P_C_* and the maximum relative capacitance change *ΔC_max_/C*_0_. (**c**) Capacitance changes (black line) and foam permittivity changes (blue line), versus applied pressure, for a sensor featuring a 6:1 small pore foam layer; (**d**) Changes of the PDMS foam thickness *d* with applied pressure for a sensor featuring a 6:1 small pore foam layer.

**Figure 5 sensors-19-01968-f005:**
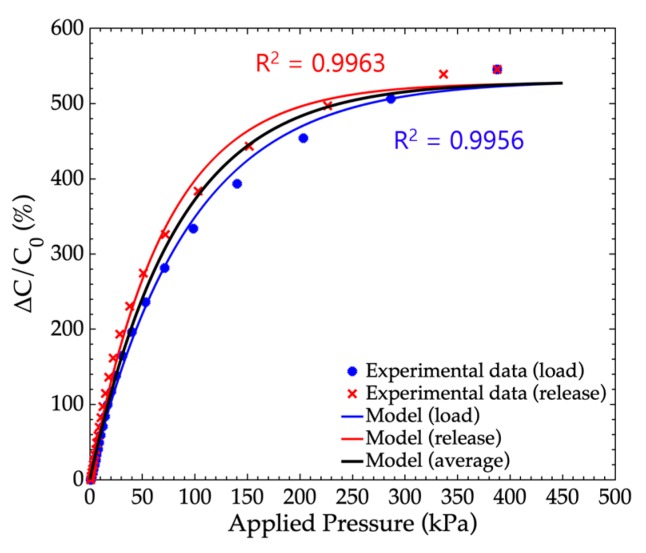
Typical response curve for a sensor featuring 4:1 small pore PDMS foam. Experimental data obtained during a load cycle (blue dots) and an adjusted load model (blue line) and a release cycle (red cross) with an adjusted release model (red line). Average load and release model is represented by the black line.

**Figure 6 sensors-19-01968-f006:**
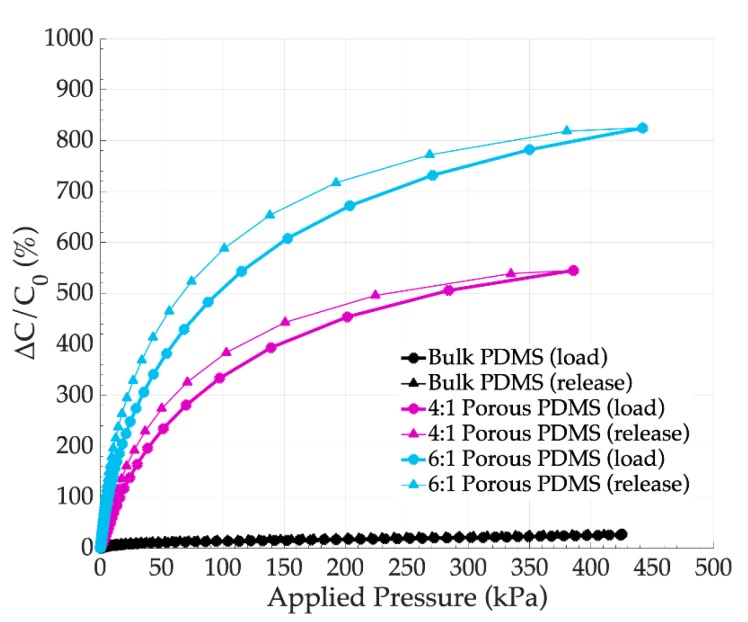
Variation of the capacitance according to the applied pressure, depending on the sugar:PDMS ratio (small pores PDMS foam, (*d*_0_ = 5 mm ± 0.5 mm).

**Figure 7 sensors-19-01968-f007:**
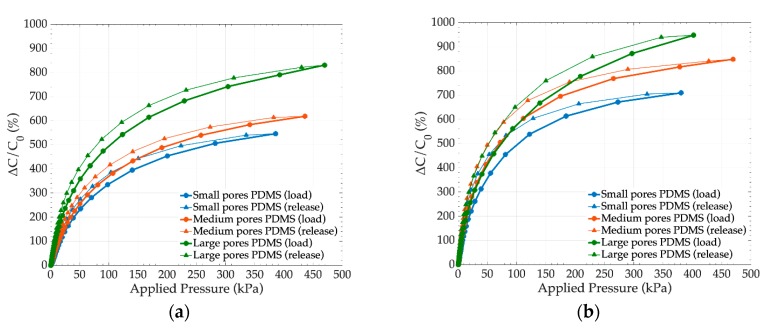
Variation of the capacitance according to the applied pressure, depending on the pore size for (**a**) 4:1 and (**b**) 6:1 PDMS foams (*d*_0_ = 5 mm ± 0.5 mm).

**Figure 8 sensors-19-01968-f008:**
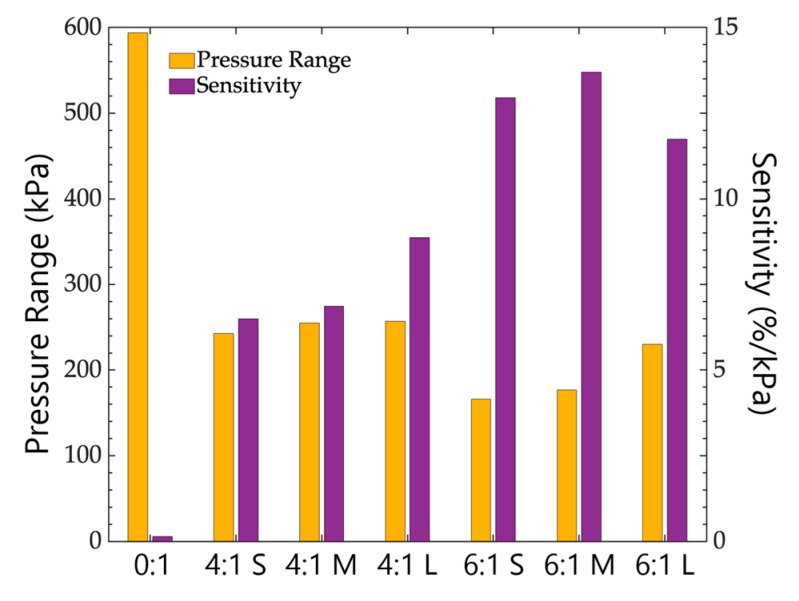
Sensitivity and pressure range of the fabricated sensors according the used bulk PDMS or PDMS foams. Sugar:PDMS ratios were 0:1, 4:1, and 6:1, and the pore sizes were small (S), medium (M), and large (L).

**Figure 9 sensors-19-01968-f009:**
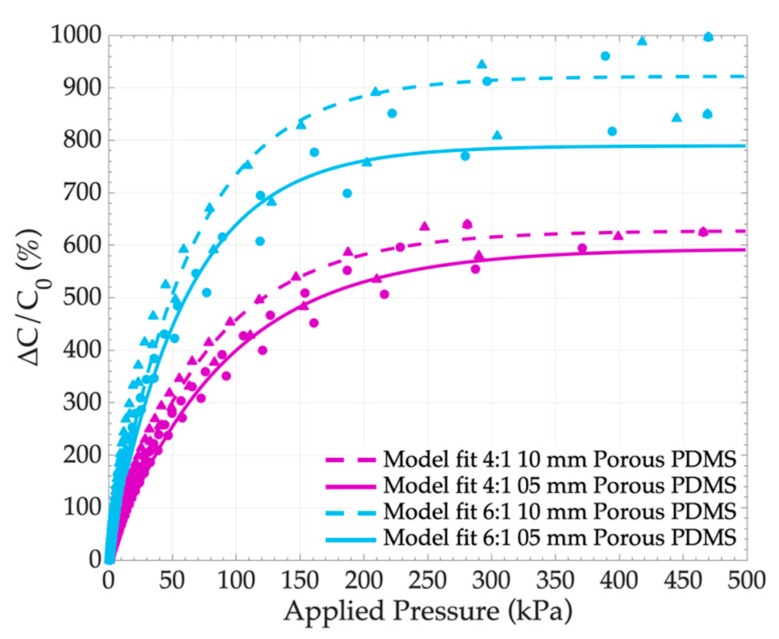
Variation of the capacitance according to the applied pressure depending on the thickness of the porous PDMS medium pores with sugar:PDMS ratios of 4:1 and 6:1.

**Figure 10 sensors-19-01968-f010:**
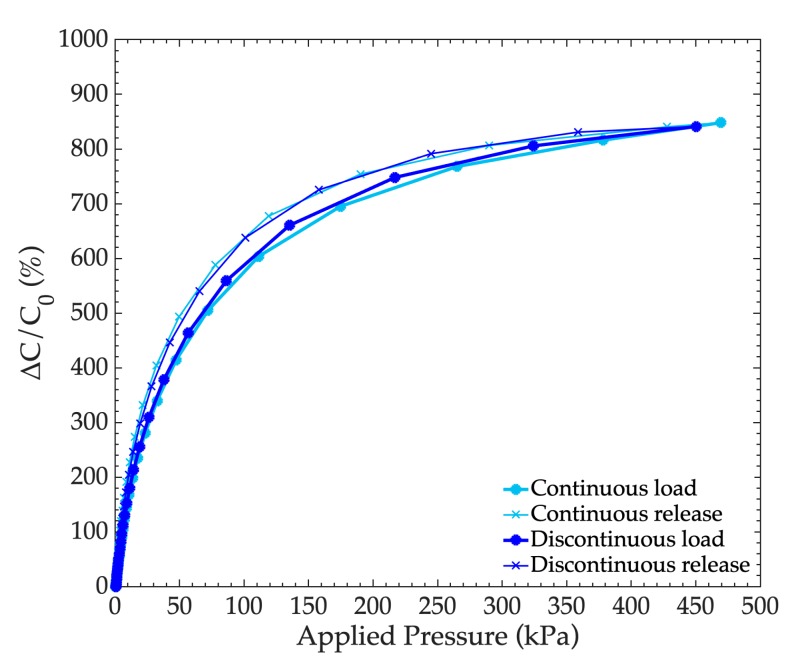
Variation of the capacitance according to the applied pressure, depending on the dynamics of the applied load (continuously or discontinuously with 10 pauses at each measurement point) for a 6:1 medium pore, porous PDMS that was 5 mm thick.

**Table 1 sensors-19-01968-t001:** Variation of the relative permittivity of the Microporous PDMS as a function of the porosity.

Volume Ratio Sugar: PDMS	Sizes of Sugar Particles (µm)	Porosity (%)	Theoretical *ε_r_*	Experimental *ε_r_*
0:1 (*bulk*)	No crystal	0	2.69	2.67 ± 0.06
4:1 (*foam*)	*Small:* 470 ± 100	78.64 ±1.40	1.36	1.56 ± 0.01
	*Medium:* 700 ± 230	80.36 ±1.40	1.33	1.36 ± 0.06
	*Large:* 1100 ± 330	80.81 ± 1.40	1.32	1.34 ± 0.01
6:1 (*foam*)	*Small:* 470 ± 100	81.96 ±1.40	1.30	1.30 ± 0.13
	*Medium:* 700 ± 230	84.36 ±1.40	1.26	1.25 ± 0.07
	*Large:* 1100 ± 330	83.53 ±1.40	1.28	1.29 ± 0.01

**Table 2 sensors-19-01968-t002:** Summary of the characterization results for PDMS bulk film and porous PDMS film as a function of the sugar/PDMS ratio and pore size (*d*_0_ = 5 mm ± 0.5 mm).

Volume Ratio Sugar: PDMS	Sizes of Sugar Particles	*C*_0_ (pF)	*PR* (kPa)	*S* (%.kPa^−1^)	Hysteresis (%)
0:1	Bulk	1.14 ± 0.02	594	0.14	8.85
4:1	Small	0.89 ± 0.02	243	6.50	5.60
	Medium	0.62 ± 0.02	255	6.87	5.41
	Large	0.54 ± 0.02	257	8.85	5.31
6:1	Small	0.82 ± 0.02	166	12.95	6.42
	Medium	0.69 ± 0.02	177	13.70	4.58
	Large	0.84 ± 0.02	230	11.74	6.62

**Table 3 sensors-19-01968-t003:** Summary of characterization results for PDMS foams as a function of the sugar:PDMS ratio and pore size, for two different thicknesses (*d*_0_ = 5 mm ± 0.5 mm and *d*_0_ = 10 mm ± 0.5 mm).

Volume Ratio Sugar: PDMS	Sizes of Sugar Particles	*d*_0_ (mm)	*C*_0_ (pF)	*PR* (kPa)	*S* (%.kPa^−1^)	Hysteresis (%)
4:1	Medium	10	0.40 ± 0.02	221	8.32	7.22
		5	0.62 ± 0.02	261	6.88	5.41
6:1	Medium	10	0.38 ± 0.02	180	15.77	5.06
		5	0.69 ± 0.02	177	13.70	4.58

**Table 4 sensors-19-01968-t004:** Summary of the characterization results for the 6:1 medium pore PDMS foam based sensor for the load applied continuously and discontinuously (*d*_0_ = 5 mm ± 0.5 mm).

Volume Ratio Sugar: PDMS	Sizes of Sugar Particles	Applied Load	*C*_0_(pF)	*PR* (kPa)	*S* (%.kPa^−1^)	Hysteresis (%)
6:1	Medium	Continuously	0.69 ± 0.02	177	13.70	4.58
		Discontinuously		174	13.66	2.92

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
