# Peer review of "Influence of the Porosity of Polymer Foams on the Performances of Capacitive Flexible Pressure Sensors†"

_sensors, 2019, doi:10.3390/s19091968_

Reviewer 1 Report

The authors present capacitive force/pressure sensors based on porous polydimethylsiloxane (PDMS) foams as compressible dielectrics. Porosity was introduced into the PDMS layer by blending differently sized sacrificial sugar crystals into the PDMS mixture and dissolving them later. The PDMS foams were sandwiched between two electrodes and the capacity change of the structure was measured. Forces compressing the foam layer were measured employing capacitance changes as sensing signal.

The present work is clearly presented. However, careful revision of the manuscript regarding grammar/spelling/style is strongly suggested.

The paper demonstrates a potent application of an interesting material. The introduced porosity has a strong effect on the sensor responses.

The manuscript can be considered for acceptance after addressing different major issues in content and form:

1) Equation 2 implies a constant value for εr. As the foams feature open pores it is to assume that during compression air will be removed, leading to a decreasing porosity f and concomitant changes in εr. Please comment on this influence on the sensor signal. If possible it would be interesting to monitor the PDMS foam thickness d during the measurement, as it would enable the observation of changes of εr. If not possible, please provide rough information regarding to which extent the foams are compressed under the given loads and an estimation of the expected capacity changes.

2) Line 65: ΔC = C0-C should be ΔC = C-C0.

3) Please clarify how the volume fraction of sugar:PDMS (4:1, 6:1) was determined. Does the volume of added sugar represent the real "sugar crystal volume" later defining the pores or the volume of the added sugar powder?

4) Please specify the chemical nature of the sugar.

5) Figure 3: Provide histograms for the pore sizes as well to document the matching of the pore sizes with respect to the particle sizes. Further, the figure caption is misleading, as the axes labels denote particle sizes rather than pore sizes. The scale bar in figure part b is barely visible.

6) The authors employ a simple exponential model for the description of the sensor response. The model is used to extract characteristics of the sensor behavior. However, the goodness of fit is limited, especially at high pressures/in the saturation range.

Theory suggests a reciprocal behavior (eq. 1, neglecting changes of εr), superimposed with the non-linear stress/strain dependence d(P) of the PDMS foam material. Please comment on the choice of the simplified model and discuss it with regard to theory.

Again, if possible, measurements of the film thickness d under load would augment the manuscript as they enable extracting the deflection behavior d(P)  for the different foams, and hence the compilation of a refined model.

Figure 5/6/8: Remove connecting lines between points and add the fit functions to the plots to assess the goodness of fit.

insets to the plots, providing magnifications of the low pressure/linear regime, would be helpful to assess the tangents' goodness of fit.

7) The left hand term of equation 5 should be ΔC/C0(P) (without the max index).

8) Inclusion of a photograph showing the PDMS foam materials/a sensor device would enhance the work.

Further some formal issues:

- carefully revise the manuscript regarding grammar and wording/spelling.

- remove "100" from equation 2; if represented in percent, ΔC/C0 is multiplied by 100%.

- remove units and the { bracket from equations in equation 6.

- don't use "camelCase" for polydimethylsiloxane.

- double-check the citation style and references, some seem to be erroneous (especially 4, 9, 12, 16).

- figures should be reproduced in higher quality/resolution, revise figure captions/data labels.

- amend the italic/non-italic typesetting of the variables' indices.

Author Response

Please find the reply to reviewer 1 in the attached file.

Reviewer 2 Report

In this work, authors present the development and study of microporous PDMS foams based of flexible capacitive pressure sensor for wearable application.

In general, the work is comprehensive, interesting. I believe that after revision of suggested changes this work would become acceptable for publication in Sensors. The suggested changes are detailed here below:

1.      In abstract Figure of merit (for e.g. sensitivity, hysteresis) regarding the sensors are missing

2.      There are many reports on capacitive pressure sensors for wearable applications.   In introduction authors need to cite recent works in this area. Please clearly explain novelties of this work and compare the performance with other similar works.  

3.      Could you please explain how you deposit the electrodes?

4.      The work is proposed for wearable applications, could you please give the information regarding influence bending on the performance of the sensors.

5.      Could you please explain is there any influence of surface area on the performance of the sensors?

6.      Could you please give a plot of sensitivity of the sensors

7.      What are the significance of this pressure ranges in wearable applications?

Author Response

Please find the reply to reviewer 2 in the attached file.

Round  2

Reviewer 1 Report

- Figure parts 4c and 4d seem to be interchanged in the respective figure caption.

- "(dis)continous" should be (dis)continuous in the legend of Figure 10

Author Response

Point 1 - Figure parts 4c and 4d seem to be interchanged in the respective figure caption.

Response to point 1 : The caption of Figure 4 has been corrected.

Point 2 - "(dis)continous" should be (dis)continuous in the legend of Figure 10

Response to point 2 : the legend in Figure 10 has been corrected.